# Unsaturated Fatty Acids and Their Immunomodulatory Properties

**DOI:** 10.3390/biology12020279

**Published:** 2023-02-09

**Authors:** Salvatore Coniglio, Maria Shumskaya, Evros Vassiliou

**Affiliations:** Department of Biological Sciences, Kean University, Union, NJ 07083, USA

**Keywords:** polyunsaturated fatty acids, inflammation, immunity, omega-3, omega-6, microglia

## Abstract

**Simple Summary:**

Diet can influence human health in both positive and negative ways. Ω-3 polyunsaturated fatty acids have an overall positive effect on humans. The sources of these fatty acids are primarily plant seeds and fish. Consumption of ω-3 fatty acids reduces inflammation and markers associated with certain diseases. Reduction in inflammation occurs through metabolites of ω-3 fatty acids and biophysical and biochemical changes in plasma membrane properties. In general, a diet high in ω-3 and low in ω-6 fats is considered favorable. This can be achieved by increasing fish and vegetable consumption while reducing animal fats in our diet.

**Abstract:**

Oils are an essential part of the human diet and are primarily derived from plant (or sometimes fish) sources. Several of them exhibit anti-inflammatory properties. Specific diets, such as Mediterranean diet, that are high in ω-3 polyunsaturated fatty acids (PUFAs) and ω-9 monounsaturated fatty acids (MUFAs) have even been shown to exert an overall positive impact on human health. One of the most widely used supplements in the developed world is fish oil, which contains high amounts of PUFAs docosahexaenoic and eicosapentaenoic acid. This review is focused on the natural sources of various polyunsaturated and monounsaturated fatty acids in the human diet, and their role as precursor molecules in immune signaling pathways. Consideration is also given to their role in CNS immunity. Recent findings from clinical trials utilizing various fatty acids or diets high in specific fatty acids are reviewed, along with the mechanisms through which fatty acids exert their anti-inflammatory properties. An overall understanding of diversity of polyunsaturated fatty acids and their role in several molecular signaling pathways is useful in formulating diets that reduce inflammation and increase longevity.

## 1. Fatty Acids in Living Systems

Fats and oils are storage lipids used across almost all living organisms [1]. Per a suggested daily ration from the Dietary Guidelines for Americans, total fat should be limited to 20–35% of daily calories [2]. Fats and oils are derivatives of fatty acids, which are carboxylic acids with hydrocarbon chains ranging in length from four to 36 carbons (Figure 1). Fatty acids (FAs) are essential components not only of fats, but also membrane lipids—phospholipids, and sphingolipids—with the latter being abundant in neural tissues [1]. In saturated fatty acids, the hydrocarbon chains contain no double bonds, and in unsaturated fatty acids they contain one or more double bonds. Simple fatty acids’ nomenclature states the number of carbons in the chain and the number of bonds after a colon; the position of the bonds is designated by Δ and a number of carbons relative to the carboxylic carbon (=α-carbon). Polyunsaturated fatty acids (PUFAs), with a double bond between a third or fourth carbon or between sixth and seventh carbon from the methyl end of the chain (=ω-carbon), are especially important for human health, and are alternatively called ω-3 and ω-6 fatty acids to reflect the fact that the physiological role of these fatty acids is more related to the position of the double bond counting from the methyl end (Figure 1).

In order to produce necessary polyunsaturated fatty acids, humans require the ω-3 PUFA α-linolenic acid (ALA, 18:3Δ^9,12,15^) but lack enzymatic ability to synthesize it, and thus must obtain it from plant sources. From this essential fatty acid, humans can then synthesize more PUFAs, such as eicosapentaenoic acid (EPA, 20:5Δ^5, 8, 11,14,17^ or ω-3 group) and docosahexaenoic acid (DHA, 22:6Δ^4,7,10,13,16,19^, also ω-3 group) [1]. EPA and DHA can be also sourced from fish and seafood [3].

## 2. Biosynthesis of Essential Fatty Acids in Plants

Biosynthesis of fatty acids in plants is located in plastids, the photosynthetic organelles of plant cells. Fatty acids are then subsequently utilized as components of plastid and endoplasmic reticulum membrane phospholipids, storage lipids, or as extracellular waxes [4]. Major plastid lipids are first synthesized using 16:0 and 18:1 acyl groups, and additional double bonds are later added by fatty acid desaturases (FADs). Desaturases insert the double bonds into fatty acid hydrocarbon chains to moderate the fluidity of lipids and corresponding membranes [5]. Membranes with phospholipids that contain saturated fatty acids are more rigid, which creates physiological issues when the membrane solidifies as a result of cold action; unsaturated fatty acids make phospholipids more flexible and allow them to resist cold stress [6,7]. Accumulation of unsaturated fatty acids such as ALA in plant membranes is a common abiotic stress response that leads to an increase in membrane fluidity and resistance to membrane rigidification caused by chilling. In addition to the moderation of membrane fluidity, C18 PUFAs act as intrinsic antioxidants. The double bonds in unsaturated fatty acids make them susceptible to reactive oxygen species (ROS) that are usually produced as a result of stress. Overexpression of ω-3 fatty acid desaturases is a method of general defense in plants, to induce stress response. In the case of an extreme increase in free radicals (which actually could be a result of photosynthesis), C18 PUFAs suffer from peroxidation, resulting in accumulation of malondialdehyde (MDA), which at low levels plays a signaling role, facilitating stress perception; however, in instances of excessive peroxidation of PUFAs, high levels of MDA and ROS may result in a massive oxidative catastrophe and DNA damage. Thus, plants maintain an accurate unsaturated fatty acid and reactive species homeostasis [8,9].

### Unsaturated Fatty Acids Synthesis

Oleic (OA, 18:1Δ^9^), linoleic (LA, 18:2Δ^9,12^), and α-linolenic (ALA, 18: 3Δ^9,12,15^) acids are key unsaturated fatty acids synthesized in plants [10]. They participate in stress responses and are precursors for plant hormones. FA synthesis is regulated by central phytohormones, abscisic acid, auxin and jasmonic acid, which organize plant growth, development and defense [11]. Hundreds of plant fatty acids, their structures and the related literature can be found using the PlantFAdb online resource [12].

Oleic acid is synthesized de novo in plastids from acetyl-coA; first, a saturated stearic acid (18:0) is produced by a fatty acid synthase (FAS) and acetyl-coA carboxylase, then stearoyl-ACP desaturase introduces a first double bond in the 9th position [10] (Figure 2). Humans can endogenously synthesize this acid and thus it is not essential [13]. As biosynthesis of C18 PUFAs in plants is coupled with synthesis of membrane lipids, oleic acid is then incorporated into a phospholipid, such as phosphatidic acid or phosphatidylcholine, for the subsequent desaturation either via prokaryotic (in chloroplasts) or eukaryotic (endoplasmic reticulum) pathways [14]. Biosynthesis of the successive linoleic and α-linolenic acids requires Δ12 (ω-6 group) and Δ15 (ω-3 group) FADs, which are present only in photosynthetic organisms. The genes for ω-6- and ω-3- FADs are found in genomes of many staple plants and in general, their expression is heavily induced by cold stress [15,16,17]. Humans lack these enzymes, thus making ω-3 and ω-6 fatty acids essential to the human diet [18].

## 3. Dietary Plant Sources of Fatty Acids

Plant oils and seeds are excellent dietary sources of essential fatty acids. ALA in high concentrations is present in flaxseed/linseed oils (Table 1). LA is found in large amounts in safflower oil. Recent genomic studies of safflower (*Carthamus tinctorius*) revealed tandem duplications of a ω-6 FAD being specifically expressed in seeds, resulting in a high content of linoleic acid [19]. In order to improve the production of 18:3 fatty acids, genetic modification has been attempted in model plants of tobacco and Arabidopsis [20,21], which resulted in improving their resistance to abiotic stress and increasing the production of 18:3 fatty acids. This can potentially lead to breeding of stress-tolerant plants with a higher nutritional quality of oils.

Oleic acid is not essential for humans, but is synthesized by many plants of nutritional value such as *Olea europaea, Brassica* species, *Arachis* species and some others (Table 1). Genome editing using CRISPR-Cas9 technique, which disrupts ω6- desaturase, has been performed in rice (*Oryza sativa*) and soybeans (*Glycine max*), resulting in oils rich in oleic acid [22,23]. High-oleic and high-stearic oils were also produced in cotton, using hairpin RNA-mediated post-transcriptional gene silencing of ω6- desaturase [24]. These high-stability cooking oils, after evaluation by food technologists, can potentially replace saturated fats and hydrogenated oils.

Many cooking oils contain a significant amount of saturated fats. For example, palmitic acid (16:0) makes up to 44% of palm oil, 26% of cocoa butter, and 8–20% of olive oil and 10–12% of soybean oil [25]. It is, however, possible to breed novel plant varieties with a higher content of PUFA and a lower content of saturated fatty acids. For example, genetic modification of soybean lines, which results in early termination of palmitoyl-acyl carrier protein thioesterase, has been shown to reduce levels of palmitic acid and can be instrumental to breeding soybean varieties with healthier oils [26].

In addition to seafood sources, plants can be instrumental in providing significant amounts of EPA and DHA. Microalgae cultivated on glucose sources as well as transgenic plants (e.g., *Camelina*) can be engineered to accumulate ω-3 EPA and DHA, presenting a sustainable alternative to fish [3,27,28].

## 4. Polyunsaturated Fatty Acids in Human Diet

The importance of food as a therapeutic tool was recognized by Hippocrates thousands of years ago with the well-known quote: “Let food by thy medicine and medicine by thy food”. It has, however, been a challenge to conclusively confirm that a particular diet can provide protection against disease. The case of PUFAs is no exception. They are capable of binding to a number of enzymes/proteins and induce a variety of effects (Table 2). A search of the PubMed database for fatty acids yields over 500,000 publications. The majority of these articles have an overall positive conclusion in terms of the benefit of unsaturated fatty acids for human health. In similar manner, in a search restricted to human clinical trials, the conclusion is also generally positive in terms of the health benefits of ω-3 polyunsaturated fatty acids [29,30,31,32,33,34]. The typical Western diet tends to be high in ω-6 polyunsaturated fatty acids due to the high consumption of meat and meat-derived products. The dominant fatty acids in meat are saturated and high in arachidonic acid (AA), an ω-6 fatty acid associated with inflammation. In general, inflammation is an immunological response seen during infection, cell injury or exposure to harmful chemical agents [35]. Furthermore, the dominance of corn and soy beans in Western diet inadvertently leads to a higher consumption of ω-6 fatty acids, as corn and soy beans are low in ω-3 fatty acids and high in ω-6. The dominant PUFA in corn and soy beans is an ω-6 linoleic acid. Seeds and their corresponding oils that are high in concentrations of ω-3 fatty acids, such as flaxseed, tend not to be common in Western diets. Any supplementation or consumption of foods high in ω-3 fatty acids favors restoration of the imbalance of ω-3 to ω-6 and likely yields health benefits.

## 5. Long Chain Polyunsaturated Fatty Acids (LC-PUFAs)

In the case of the long chain polyunsaturated ω-3 fatty acids (LC-PUFAs), docosahexaenoic (DHA) and eicosapentaenoic acid (EPA) are the most common and widely used as supplements. Common sources include fish oil, krill oil and ethyl esters. Consumption of fish yields significant health benefits to humans and is believed to be partially responsible for the health benefits of the Mediterranean diet [33,36,37]. Krill oil consumption leads to the highest incorporation of DHA and EPA into plasma phospholipids [38]. The ability of the human body to absorb DHA and EPA is also affected by the environment of the fatty acid [39]. Consumption of intact salmon yields a higher absorption and bioavailability when compared to fish oil. Nonetheless, ω-3 fatty acids have been shown to be incorporated into plasma membranes regardless of the source [40]. The mere substitution of saturated fatty acids with PUFAs in a diet increases bacterial family populations of *Lachnospiraceae* and *Bifidobacterium* spp. [41]. This phenomenon brings a different dimension with respect to the health benefits of PUFAs in the human diet. If alterations in specific microbiota populations can impart a positive effect on the hosts, PUFAs are undoubtedly capable of such an effect. It is striking that certain bacterial populations are associated with lower inflammation and that diet indirectly can exert anti-inflammatory properties [31,42,43]. Consumption of walnut, an excellent source of ALA, has been shown to alter the microbiota bacterial family *Lachnospiraceae*. While fatty acid supplementation in the form of oil capsules is very common and can convey similar benefits, it seems more advantageous to consume whole foods rich in specific PUFAs. Examples of such foods include salmon, sardines, walnuts and flaxseeds. Algal sources offer an attractive alternative to fish whose DHA content can vary depending on whether they are farmed or wild caught. The oxidative stability of fatty acids during digestion is another parameter that still requires further investigation. The nature of fatty acids makes them susceptible to peroxidation during digestion [44]. A considerable amount of peroxidation occurs in the gastric phase of digestion. Emulsification and encapsulation can provide protection in the gastric phase, and ultimately increase the bioaccessibility of PUFAs [45]. Food preservation involving smoking of salmon oils has been shown to reduce peroxidation of salmon oils, and provides a relatively low-tech means of preventing peroxidation of PUFAs in fish [46].

## 6. Human Clinical Trials Involving Polyunsaturated Fatty Acids

Numerous clinical trials involving DHA and EPA supplementation show an overall positive impact on a wide range of diseases such as Crohn’s disease, major depressive disorder, cardiovascular disease, autism, hypertension, arthritis and lupus [30,32,40,47,48,49,50,51]. It must be noted though that a significant number of clinical trials do not show any measurable health benefits [52,53,54,55]. Conflicting reports often lead to doubts as to the efficacy of PUFAs on human health. Single nucleotide polymorphisms (SNPs) at the level of the fatty acid desaturases 1, 2, 3 (FADs 1,2,3) and elongases (ELOVL 2,5) have been reported to interact with the overall production of PUFAs in the colostrum of pregnant women [56]. It is therefore reasonable to assume that the genetic makeup of individuals can have a positive or negative interaction on PUFA supplementation, and can partly explain why several studies yield conflicting results. In a study with Danish infants who were homozygous for the FADs minor allele rs1535, there was a 1.8% DHA increase. On the contrary, minor allele carriers of the rs174448 and rs174575, had a DHA reduction [57]. Interestingly, breastfeeding duration had a positive impact on DHA levels in these infants, regardless of genotype. In a study conducted in Mexico, the maternal rs174602 SNP had a positive enrichment on infant amino acid and amino sugar metabolic pathways, and decreased fatty acid metabolism [58]. Supplementation with EPA in females has been shown to result in higher plasma DHA levels when compared to males [59]. Polymorphism rs953413 of the ELOVL2 gene seems to exert an influence on DHA plasma levels. Collectively, these clinical data support the use of a genetic analysis of participants of studies involving long chain PUFA supplementation, particularly of the FADs and ELOVL genes.

ALA, a shorter ω-3 PUFA, has comparatively less pronounced health benefits with regard to DHA and EPA [60]. Despite supplementation of 67 healthy individuals with 3.6 g/day of ALA for 8 weeks, no significant improvements were observed in terms of oxidative stress, inflammation and blood pressure. In a study involving 59 untreated pre-hypertensive patients supplemented with 4.7 g/day ALA, a reduction in TNF-α and free fatty acids was observed, but not in any other vascular markers [61].

Oleic acid, which is found in high concentrations in olive oil and peanuts, is an ω-9 monounsaturated fatty acid. Consumption of olive oil and peanut oil has been associated with lower risk of developing asthma and improved glucose regulation [62,63]. However, in a study involving patients with stable coronary disease, intake of either extra virgin olive oil or pecans had no effect on plasma fatty acids [64]. In another study investigating inflammatory markers and oxidative status in obese men, no measurable reduction was observed after consuming either high oleic or conventional oleic peanuts [65].

Ω-6 polyunsaturated fatty acids such as LA and AA are considered less desirable in terms of health benefits. The overall consensus is that the ratio of ω-3 to ω-6 should be as high as possible. It is believed that humans evolved with an ω-3 to ω-6 ratio of 1 to 1, but currently Western diet is around 1 to 15 [66].

## 7. Anti-Inflammatory Properties of Polyunsaturated Fatty Acids

Ω-3 LC-PUFAs (DHA and EPA) tend to have a more consistent anti-inflammatory effect on immune cells in comparison to other PUFAs [63,67,68]. The anti-inflammatory effect is observed both in vitro and in vivo [49,69,70]. There are several pathways that have been elucidated which provide evidence as to how fatty acids induce an anti-inflammatory state. The peroxisome proliferator-activated receptors (PPARs) have been shown to bind to DHA and EPA and ultimately suppress the production of cytokines related to the NF-κB inflammatory master transcription factor. Another pathway through which fatty acids reduce inflammation is through resolvins and neuroprotectins [71,72]. Aspirin acetylation of the COX enzyme significantly increases production of resolvins and neuroprotectins in the presence of DHA and EPA. Fatty acids incorporated in the plasma membranes of cells are all susceptible to cleaving by phospholipases during inflammatory stress. AA is known to be involved in the production of various prostanoids, via the COX pathway, that have an overall inflammatory effect [73]. The proportion of AA to DHA and EPA is thought to affect inflammation. Specifically, in the presence of aspirin and subsequent acetylation of the COX-2 enzyme, production of resolvins is enhanced, thereby reducing the impact of the inflammatory prostaglandins generated by AA. Another mechanism through which PUFAs can exert anti-inflammation is by affecting the plasma membrane’s properties through lipid rafts. Lipid rafts are domains in plasma membranes that are characterized by higher levels of cholesterol, glycophospholipids and receptors [74]. They play a key role in several cellular activities, including endocytosis, cell signaling and exocytosis. DHA was shown to decrease levels of lipid rafts by as much as 30%. In cancer cells, this reduction in lipid rafts was associated with a reduction in cell proliferation. Both DHA and EPA incorporation increased levels of the antigen presenting molecule MHC I that is expressed on all nucleated cells [75]. Efficient antigen presentation via MHC I leads to faster immune resolution and reduction of chronic inflammatory conditions which are thought to support malignant cell proliferation. The increase in MHC I expression was not attributable to any conformational changes affecting antibody binding to MHC I, but rather to an increase in the plasma membrane. Surprisingly, even AA exhibits an immunomodulatory effect in some instances by preventing M2 polarization of macrophages [76]. Opposing this effect is PGE2, a by-product of AA, which enhances M2 polarization. In another study, AA clearly aggravated obesity and increased inflammatory microbiota [77]. Once again, the balance of ω-3 to ω-6 fatty acids seems to be the key factor in attaining an anti-inflammatory state.

### 7.1. PUFAs and Neuroinflammation—Effect on Brain Microglia

There is a substantial body of literature addressing the role of PUFAs in neuroinflammation [78]. The brain is a lipid-rich organ containing a great diversity of lipid species, especially PUFAs. The main source of PUFA in the brain is derived from the diet and needs to enter the brain [79]. Lipids in general have three primary mechanisms of entry into the brain across the blood–brain barrier (BBB): passive diffusion, transcytosis via receptor-mediated endocytic pathways, and transport using transmembrane proteins [80]. Conditions that disrupt neuronal homeostasis result in brain synthesized PUFAs. For example, bacterial endotoxin lipopolysaccharide (LPS, an inflammatory stimulus) treatment of astrocytes results in upregulated synthesis of AA and DHA [81].

Microglia are the resident immune cell type of the CNS, and make up roughly 15% of glial cells in the brain. Their hallmark feature is a branched and ramified morphology which facilitates sensing and response to brain injury and infection. Most studies which address neuroinflammation focus on the status of microglia, as they play a central role in most neurological disorders including Alzheimer’s disease (AD), multiple sclerosis (MS), ischemic stroke, traumatic brain injury (TBI) and even cancer. Microglia, like other tissue macrophages, can be polarized into “classical” pro-inflammatory phenotype and the “alternative” anti-inflammatory phenotype, often referred to as M1 and M2, respectively, although these likely represent extremes within a spectrum of responses. The M1 state is typically activated by TLR ligands such as LPS which induce the expression of cytokines such as TNF-α, IL-1β and IL-6. The M2 phenotype results from stimulation with cytokines such as IL-4 and IL-13, and promotes expression of cytokines and growth factors such as IL-10 and TGFb, which are immunosuppressive and promote tissue regeneration. With respect to neurological disorders, microglial expression of M1 markers correlates with poorer clinical outcomes. In multiple animal models, experimental manipulation, which produces M1 microglia (such as LPS injection), generally exacerbates the disease severity [82,83,84]. Conversely, the repolarization of microglia to the M2 anti-inflammatory state tends to have a beneficial effect. Therefore, therapeutic intervention which promotes the M2 polarization of microglia would be desirable for many neurological diseases. In this regard, PUFAs have gained attention as potential modulators of neuroinflammation.

In general, saturated FAs induce a pro-inflammatory phenotype in microglia, while MUFAs and PUFAs promote the M2 state. Saturated PA has been shown to stimulate expression of pro-inflammatory cytokine gene expression to a similar extent as LPS in cultured astrocytes and BV-2 microglial cells in a TLR4-dependent manner [85,86], whereas unsaturated OA has been shown to have the opposite effect [87]. In an in vitro model, it was recently shown that OA can mitigate the effects of PA-stimulated microglia neurotoxic effects in neuronal cocultures [88]. At the forefront of the role of lipids in neuroinflammation are the ω-3 long chain PUFAs and their derivatives, specialized pro-resolving lipid mediators (SPMs). This class of lipids have emerged as central players in limiting the M1 phenotype and dampening neuroinflammation. It is well established that ω-3 long chain PUFAs downregulate LPS-stimulated pro-inflammatory genes such as TNF-α and IL-6 in microglia, both in vitro and in vivo [87,89,90,91,92]. DHA is a potent M2 polarizing agent in microglia, and can reduce inflammation in several neuronal disease models [93,94,95]. Furthermore, mice deficient in DHA synthesis exhibit increased M1 inflammatory markers in the brain [96]. The ω-3 long chain PUFAs were discovered to work on multiple levels, including regulation of receptor activity, signaling kinases and gene expression. ALA, which is found in walnut extract, downregulates the expression of surface TLR4 and iNOS induction [97,98,99]. DHA and EPA were shown to enhance SIRT-1 deacetylase activity, which had the effect of blocking NF-kB activation of inflammatory genes in the MG6 murine microglial cell line [100]. The effect of DHA on BV-2 microglial cells was recently analyzed in depth using quantitative proteomics [101]. Confirming earlier studies, many signaling proteins in the NF-kB pathway, including sequestome-1, NOS and CD40 were found to be differentially expressed [102]. Interestingly, it was also discovered that DHA influences the pattern of protein expression involved in fatty acid metabolism and ribosome function, suggesting a more global mechanism of DHA regulation of microglial function. Much of the anti-inflammatory activity these dietary PUFAs is likely mediated by SPMs. One of the SPMs, resolvin RvD1, can enhance IL-4 induced M2 polarity of BV-2 microglia in vitro [103]. RvD1 can also promote this shift away from M1 in vivo [104].

### 7.2. PUFAs and Neuroinflammation—Effect on Brain Astrocytes

Astrocytes are the main glial cells of the brain and play essential roles in energy metabolism, maintenance of extracellular ion concentrations, formation of the blood–brain barrier and general CNS homeostasis. Astrocytes are also able to modulate inflammation both directly via release of soluble mediators and indirectly by influencing neighboring microglia [105,106]. Astrocytes can also respond to signals from damaged neurons, resulting in reactive astrogliosis, which is characteristic of neuroinflammation in general. LPS stimulation of astrocytes results in a release of DHA which likely promotes survival of neurons during neuroinflammation [81]. PUFAs can also act on astrocytes to attenuate inflammation. DHA and EPA-treated astrocytes show a decrease in NF-kB and immunoproteosome activity [107]. Fortasyn Connect©, a nutrient combination which includes DHA and EPA, is able to inhibit astrogliosis in an in vitro model [108].

## 8. PUFAs and Neurological Diseases

Alzheimer’s disease (AD) in an invariably fatal disease marked by a steady decline in cognitive abilities and neurodegeneration. The role for microglia in AD has recently been highlighted, as they carry out phagocytic removal of amyloid plaques [109,110]. The potential for using ω-3 long chain PUFAs to ameliorate AD symptoms and progression has gained much attention [111]. In a variety of animal models for AD, diets and formulations rich in ω-3 long chain PUFAs had anti-inflammatory effects and a positive impact on disease progression [112,113,114,115,116,117]. Amyloid aggregates have the ability to promote M1 neuroinflammation, and this is dampened in the presence of DHA [118]. EPA can protect against an amyloid injection AD model in rats [119,120,121]. Interestingly, DHA and EPA treatment was shown to simultaneously enhance phagocytosis of amyloid protein while promoting an M2 phenotype [122,123]. EPA was also shown to enhance neuroprotective factors produced by astrocytes in the rat hippocampus [93]. ALA can also mobilize microglia to phagocytose extracellular tau aggregates [124]. ALA has also been shown to act via astrocytes, as conditioned media collected from human astrocyte cultures protected SH-SY5Y cells from amyloid-induced cell death [125]. Fish oil was also shown to activate AQP4 on astrocytes, resulting in glymphatic clearance of amyloid from the brain [126]. SPMs and the pathways they govern are also being tested for their ability to treat AD in preclinical models. AD patients exhibit fewer SPMs in the hippocampus, and the maresin MaR1 can inhibit M1 gene expression in the human microglial cell line CHME3 [127]. MaR1 is also able to reduce amyloid induced cytokine stimulation, alongside inducing amyloid phagocytosis by macrophages in vitro [128]. MaR1 can prevent microglia, and astrocyte activation in an AD mouse model had substantial benefits in preventing cognitive decline as well [129]. Administration of resolvin RvE1 and lipoxin LXA4 into the intraperitoneum of 5XFAD mice decreased neuroinflammation and lowered the amyloid plaque burden [130]. Intranasal delivery of a mixture of SPMs including resolvins RvE1, RvD1, RvD2, maresin MaR1 and neuroprotectin D1 was able to reduce microglial activation and restore some brain function in an AD mouse model [131].

In addition to AD, the immunomodulatory effects of PUFAs have been shown to have beneficial effects in experimental models for several other neurological diseases, including multiple sclerosis (MS), traumatic brain injury (TBI), stroke and Parkinson’s disease (PD). In general, microglial activation and expression of M1 inflammatory genes tends to contribute to neuronal cell death. Conditioned media from THP-1 macrophages treated with PA, but not OA or LA, induces apoptosis of SH-SY5Y neuronal-like cells in culture [132]. Similarly, DHA treatment of LPS-activated BV-2 translates into a less cytotoxic effect on neurons in cell cocultures [91]. In a model of myelin-induced damage which approximates pathological features of multiple sclerosis (MS), DHA and EPA shift the microglial response away from M1 [133]. Metabolites of DHA are lower in the chronic model of MS [134]. Furthermore, in the experimental autoimmune encephalomyelitis model of MS, a triglyceride formulation of DHA is able to prevent inflammation and exert neuroprotective effects [135]. There is also evidence for PUFAs in neuroprotection from stroke and brain injury. Consistent with the studies cited above, ω-3 long chain PUFAs are able to prevent neuroinflammation associated with ischemic damage and promote an M2 phenotype in microglia associated with at the site of injury [136,137]. There are multiple reports which show that DHA, either as a single agent therapy or prepared in dietary formulations, can prevent inflammation and neuronal damage in a mouse model [138,139,140,141,142,143]. DHA treatment of rats with traumatic brain injury showed less microglial endoplasmic reticulum stress and autophagy [144,145]. The mechanism of action is in part via the ω-3 fatty acid receptor GPR120 [146]. SPMs are also likely to play a role in preventing neuroinflammation associated with neuronal damage [147]. These results are encouraging and have spurred the development of small molecule agonists of SPM receptors to treat neurological disorders involving brain injury [148].

**Table 1 biology-12-00279-t001:** Content of important fatty acids in plant oils commonly used as food sources (% from all FA) [149,150,151,152].

							Rich in ω-3 and ω-6		
	Sunflower Oil	Rapeseed Oil	Mustard Oil	Peanut Oil	Olive Oil	Avocado Oil	Grapeseed Oil	Flaxseed Oil	Walnut Oil
Palmitic acid (PA)	5.94	3.97	10.24	9.37	15.11	10.08	7.2	5.87	6.3
Oleic acid (OA)	30–80 *	63.68	36.65	55.33	68.85	60.7	19.9	17.41	20.5
Linoleic acid (LA, ω-6)	21–70 *	17.43	22.06	23.69	8.5	11.8	68.1	15.76	55.5
α-linolenic acid (ALA, ω-3)	0.79	-	-	-	0.54	1.2	0.1	55.40	14.8

* Content depends on a plant breed and industrial processing.

**Table 2 biology-12-00279-t002:** PUFAs and their relevant metabolic enzymes/ligand targets.

PUFA	Transcription Factor/Enzyme	Metabolite/ Ligand	Inflammatory Effect
**Docosahexaenoic acid (DHA)**	COX-2	Resolvin D	Anti [71]
	PLA_2_	Protectin D1	Anti [153]
	PPAR-α	Ligand	Anti [154]
	PPAR-γ	Ligand	Anti [154]
**Eicosapentaenoic acid (EPA)**	COX-2 PPAR-α PPAR-γ	Resolvins E Ligand Ligand	Anti [72] Anti [155] Anti [156]
**α-Linolenic acid (ALA)**	PPAR-α PPAR-γ	Ligand Ligand	Anti [157] Anti [158]
**Arachidonic acid (AA)**	COX-2 COX-2 PPAR-α PPAR-δ	PGE_2_ PGI_2_ Ligand Ligand	Pro [159] Anti [160] Anti [157] Anti-Apoptotic [161]
**Linoleic acid (LA)**	PPAR-α	Ligand	Energy Control [157,162]
**Oleic acid (OA) ***	TLX-NR2E1	Ligand	Neurogenesis, Anti [157,163]

* MUFA, monounsaturated fatty acid.

## 9. Conclusions

The anti-inflammatory properties of unsaturated fatty acids have been shown in multiple studies. PUFAs from the ω-3 group reduce inflammation in multiple tissues, including neural tissues. Increasing ω-3 PUFAs and ω-9 MUFAs in the human diet through enrichment with plant food sources such as flaxseed/linseed, walnut oil, olive oil, and fish products may be beneficial in decreasing the overall inflammatory response in the human body. As we gain more knowledge in terms of our understanding of how various PUFAs interact with each other, it is possible that improved formulations will arise. A better understanding of how individual genotypes influence the absorption and metabolism of PUFAs will also help to design better studies utilizing PUFAs. Lastly, alternative routes of administration, other than the typical oral administration of PUFAs, may yield more pronounced health benefits in the future.

## Figures and Tables

**Figure 1 biology-12-00279-f001:**
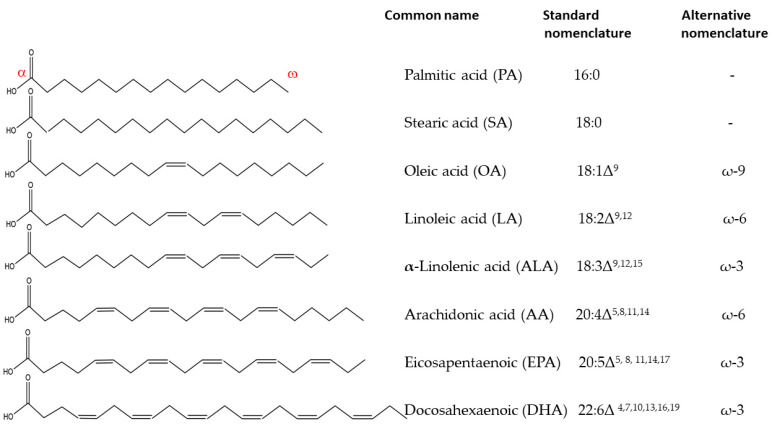
Fatty acids important for human health.

**Figure 2 biology-12-00279-f002:**
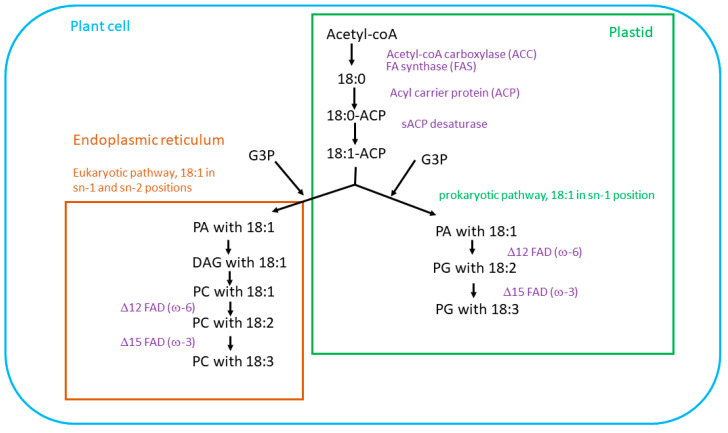
An oversimplified scheme of plant fatty acid biosynthesis. Phopspholipids PA and PC are shown as examples. G3P—glycerol-3-phosphate, PA—phosphatidic acid, PG—phosphatidyl glycerol, PC—phosphatidylcholine, DAG—diacylglycerol.

## Data Availability

Not applicable.

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
