# Peer review of "Unsaturated Fatty Acids and Their Immunomodulatory Properties"

_biology, 2023, doi:10.3390/biology12020279_

Round 1

Reviewer 1 Report

Reviewers' comments:

In the present review paper, Coniglio et al. summarized the functional role of ω-3 polyunsaturated fatty acids in immune signaling pathways in both peripheral and CNS immunity. The authors also discussed the clinical studies applying the ω-3 polyunsaturated fatty acids. However, there are several issues that the authors would like to address to improve the manuscript and they are as follows:

1. Given that astrocyte is another crucial cell type in mediating the innate and adaptive immune responses in the central nervous system. Authors should give a detailed literature review on ω-3 fatty acid on astrocytes as a separate section from 7.1.

2. According to the abstract, ω-3 polyunsaturated fatty acids in CNS disease should be the major section of the paper. However, the author just discussed the AD in section 8. The author should also include the other major CNS disease types, especially Parkinson’s disease, stroke, meningitis, multiple sclerosis, pain, and brain tumor, and give a detailed literature review.

3. Accumulating evidence showed that ω-3 polyunsaturated fatty acids exert their biological activities mainly through the formation of bioactive lipid metabolites. The author should have a detailed literature review on major types of lipid metabolites and discuss their biological functions in CNS diseases.

Author Response

REVIEWER #1:

In the present review paper, Coniglio et al. summarized the functional role of ω-3 polyunsaturated fatty acids in immune signaling pathways in both peripheral and CNS immunity. The authors also discussed the clinical studies applying the ω-3 polyunsaturated fatty acids. However, there are several issues that the authors would like to address to improve the manuscript and they are as follows:

Q1: Given that astrocyte is another crucial cell type in mediating the innate and adaptive immune responses in the central nervous system. Authors should give a detailed literature review on ω-3 fatty acid on astrocytes as a separate section from 7.1.

A1: We have added a section that discusses the role of ω-3 polyunsaturated fatty acids  in astrocyte control of  neuroinflammation with particular emphasis on neuroprotection and Alzheimers Disease. 

Q2. According to the abstract, ω-3 polyunsaturated fatty acids in CNS disease should be the major section of the paper. However, the author just discussed the AD in section 8. The author should also include the other major CNS disease types, especially Parkinson’s disease, stroke, meningitis, multiple sclerosis, pain, and brain tumor, and give a detailed literature review.

A2: We have modified the abstract so that it is clear that the review involves the role of PUFAs with respect primarily to the immune system. The CNS section is intended to be a component, not the major section. The way it was worded, as the reviewer correctly pointed out, is that CNS is the major section of the review; instead, we would rather want it to be an additional section without many details.

Q3. Accumulating evidence showed that ω-3 polyunsaturated fatty acids exert their biological activities mainly through the formation of bioactive lipid metabolites. The author should have a detailed literature review on major types of lipid metabolites and discuss their biological functions in CNS diseases.

A3: We have included more references to specialized pro-resolving mediators (SPMs) and their roles in specific neurological diseases.

Reviewer 2 Report

The paper “Polyunsaturated Fatty Acids and their Immunomodulatory Properties” describes well the functionality of different PUFA’s in controlling inflammation in cells and tissues such as neural tissues or for other PUFA’s enhancing inflammation. The potential for controlling neurological disease with food or supplements with PUFA’s such as DHAEPA and ALA is described and the scientific literature on the subject given.

What is missing was the effect of PUFA’s oxidation that can occur readily in food’s preparation and indigestion and which has toxic effects opposite of the health effects. This should mentioned as well as the literature on the subject given.

Author Response

REVIEWER #2

  the functionality of different PUFA’s in controlling inflammation in cells and tissues such as neural tissues or for other PUFA’s enhancing inflammation. The potential for controlling neurological disease with food or supplements with PUFA’s such as DHAEPA and ALA is described and the scientific literature on the subject given.

Q4. What is missing was the effect of PUFA’s oxidation that can occur readily in food’s preparation and indigestion and which has toxic effects opposite of the health effects. This should mentioned as well as the literature on the subject given

A4: Relevant information has been included regarding peroxidation during digestion and food preparation. 

Reviewer 3 Report

General comment:

This manuscript aims to present the immunomodulatory properties of polyunsaturated fatty acids (PUFAs). Overall, it is interesting and well written. However, there is some confusion regarding the different types of fatty acids and their effects. For example, the authors repeatedly speak about oleic acid, which is not a PUFA (eg: Row 259), and mention the antioxidant properties of PUFAs without always making a clear distinction between omega-3 and omega-6 PUFAs (eg: Row 206). The title of the manuscript should therefore be rather “Unsaturated fatty acids…” and not “Polyunsaturated fatty acids…”. Moreover, the link between certain PUFA-modulated pathways and inflammation is not always presented clearly (eg: Row 224). There is some confusion between anti- and pro-inflammatory effects. In this manuscript, the anti-inflammatory effects are mainly attributed to omega-3 PUFAs. Unless I am mistaken, there is no reference showing that the binding of omega-6 PUFAs on PPAR has an anti-inflammatory effect, as suggested in table 2.

Detailed comments:

Row 66: The principle that C18 PUFAs are "intrinsic antioxidants" is not very clearly explained here, because it is said that their peroxidation produces an accumulation of malondialdehyde which causes damage from oxidation. Do they act as antioxidants or prooxidants?

Rows 72 and 84: The use of abbreviations is not always respected in the text. Alpha-linolenic acid should be rated ALA. Please check the use of abbreviations throughout the text.

Rows 78 – 81: A figure showing the different steps of essential fatty acid biosynthesis in plants would make the article easier to read.

Rows 123 and 162: “Poly unsaturated” should be “polyunsaturated”.

Row 124: The abbreviation “PUFAs” should then be used everywhere to replace polyunsaturated fatty acids.

Row 127: It is unclear whether this review includes monounsaturated fatty acids or speaks only about polyunsaturated fatty acids. Should the title be changed as “Unsaturated fatty acids and their immunomodulatory properties”?

Row 137: Linoleic acid should be abbreviated as LA.

Row 144: It is surprising to read that ethyl esters are common nutritional sources of omega-3 PUFAs like fish and krill oil? Please explain!

Row 162: Algae should be mentioned, as well as that the level of omega-3 PUFAs in fish can vary widely depending on where they come from, fish caught at sea or farmed.

Row 178: The impact of breastfeeding on DHA levels is not very well explained (increase or decrease?).

Row 200: Is there an optimal ratio as omega-3 / omega-6 (1:1, 1:2, or 1:4) ?

Row 204: As substrates for prostaglandins synthesis from COX-2, other PUFAs (omega-6) are expected to be pro-inflammatory rather than anti-inflammatory, even if they are PPAR-alpha ligands. Please, rephrase!

Row 207: According to Table 2, not only DHA and EPA but also ALA, AA, and LA bind to PPAR.

Row 212: The immunomodulatory effect of EPA and DHA with respect to prostaglandin synthesis from AA (competitive synthesis between omega-3 and omega-6) is not well explained. The term immunomodulation is referring precisely to this mechanism.

Row 221: What is the link between cancer cell proliferation and inflammation?

Row 222: How can DHA and EPA have an anti-inflammatory effect by increasing the level of MHC I on cell surface? Rather, it is expected to trigger an inflammatory response?! Please explain!

Row 226: This paragraph speaks almost exclusively of the effects of DHA. What are the effects of AA (omega-6)?

Figures and Tables:

Table 1: As the review talks about polyunsaturated acids, it would be useful to subdivide Table 1 into oils rich in omega 3 and 6.

Table 1: Nutritionally, flaxseed oil and linseed oil are the same. In my opinion, it is not necessary to present the fatty acid content of both oils in the table. Flaxseed oil could be replaced advantageously by walnut oil, another oil rich in omega 3, as mentioned on row 158.

Table 2: It may be useful to differentiate COX-1 from inducible COX-2.

Author Response

REVIEWER #3

This manuscript aims to present the immunomodulatory properties of polyunsaturated fatty acids (PUFAs). Overall, it is interesting and well written. However, there is some confusion regarding the different types of fatty acids and their effects. 

Q5: For example, the authors repeatedly speak about oleic acid, which is not a PUFA (eg: Row 259), and mention the antioxidant properties of PUFAs without always making a clear distinction between omega-3 and omega-6 PUFAs (eg: Row 206). The title of the manuscript should therefore be rather “Unsaturated fatty acids…” and not “Polyunsaturated fatty acids…”. 

A5: Title was revised as per reviewer suggestion. 

Q6: Moreover, the link between certain PUFA-modulated pathways and inflammation is not always presented clearly (eg: Row 224). There is some confusion between anti- and pro-inflammatory effects. In this manuscript, the anti-inflammatory effects are mainly attributed to omega-3 PUFAs. Unless I am mistaken, there is no reference showing that the binding of omega-6 PUFAs on PPAR has an anti-inflammatory effect, as suggested in table 2.

A6: Binding of omega-6 (AA and LA) to PPAR does not have anti-inflammatory properties. The anti-inflammatory designation refers to the metabolites of omega-6 (AA and its metabolites PGI2 and PGE2). PPAR binding references show anti-apoptotic and energy control. 

Q7: Row 66: The principle that C18 PUFAs are "intrinsic antioxidants" is not very clearly explained here, because it is said that their peroxidation produces an accumulation of malondialdehyde which causes damage from oxidation. Do they act as antioxidants or prooxidants?

A7: Thank you for this valuable comment, we have fixed the sentence so it is more clear. Formation of malonidaldehyde from UFA is a result of fatty acid peroxidation due to stress; however, in low amounts MDA will work as a signal of stress, and in high amounts it would contribute to stress-related oxidation development; thus plants must maintain a careful balance of unsaturated fatty acids. It is common in plants to respond not only to a chemical molecule or hormone specifically, but also to react to its concentration and sometimes in a totally different way, and the molecule can act differently in different tissues. For example, in a famous example of auxin, high concentration of auxin stimulates stem growth but inhibits root growth. This is a very interesting topic, but we did not go into the details since our paper does not discuss plant biochemistry, and we mention plants just as a source of UFA.

Q8:Rows 72 and 84: The use of abbreviations is not always respected in the text. Alpha-linolenic acid should be rated ALA. Please check the use of abbreviations throughout the text.

A8: We have added the abbreviation for ALA  to the (ex) line 72; we will add abbreviations throughout. However, we would like to keep full names of fatty acids occasionally, to keep the text more alive and not overloaded with various abbreviations.

Q9: Rows 78 – 81: A figure showing the different steps of essential fatty acid biosynthesis in plants would make the article easier to read.

A9: Actually, biosynthesis of fatty acids in plants is a pretty complicated process and is very simplified in the text, since the manuscript is about inflammation and UFA, not plant biosynthesis. We have inserted a simplified figure. 

Q10:Rows 123 and 162: “Poly unsaturated” should be “polyunsaturated”.

A10: We fixed this typo across the manuscript.

Q11:Row 124: The abbreviation “PUFAs” should then be used everywhere to replace polyunsaturated fatty acids.

A11: We thank the reviewer for this suggestion and occasionally replaced the full name with the abbreviation. However, we would like to keep the spelled out version here and there to keep the language more alive. 

Q12: Row 127: It is unclear whether this review includes monounsaturated fatty acids or speaks only about polyunsaturated fatty acids. Should the title be changed as “Unsaturated fatty acids and their immunomodulatory properties”?

A12: This is a better title, we have changed it to the one suggested by the reviewer.

Q13: Row 137: Linoleic acid should be abbreviated as LA.

A13: We changed it and decided to keep the spelled out version instead. The sentences around contain a lot of w3 and w6 abbreviations and another abbreviation would complicate the text.

Q14: Row 144: It is surprising to read that ethyl esters are common nutritional sources of omega-3 PUFAs like fish and krill oil? Please explain!

A14: Ethyl esters were removed as common nutritional sources, EPA Ethyl ester is the most common prescription (Vascepa). 

Q15: Row 162: Algae should be mentioned, as well as that the level of omega-3 PUFAs in fish can vary widely depending on where they come from, fish caught at sea or farmed.

A15:Algae and farmed vs wild fish DHA content included.

Q16:Row 178: The impact of breastfeeding on DHA levels is not very well explained (increase or decrease?).

A16: Clarification included to show that duration of breastfeeding resulted in higher DHA levels despite the fact specific genotypes could influence levels of DHA. 

Q17: Row 200: Is there an optimal ratio as omega-3 / omega-6 (1:1, 1:2, or 1:4) ?

A17: Information regarding optimal evolutionary ratio and typical Western diet ratio included. 

Q18: Row 204: As substrates for prostaglandins synthesis from COX-2, other PUFAs (omega-6) are expected to be pro-inflammatory rather than anti-inflammatory, even if they are PPAR-alpha ligands. Please, rephrase!

A18: Modified: .Arachidonic acid is known to be involved in the production of various prostanoids via the COX pathway that have an overall inflammatory effect. 

Q19: Row 207: According to Table 2, not only DHA and EPA but also ALA, AA, and LA bind to PPAR.

A19: Indeed this is the case, however unlike ω-3 fatty acids, their anti-inflammatory effect is not clear. 

Q20:Row 212: The immunomodulatory effect of EPA and DHA with respect to prostaglandin synthesis from AA (competitive synthesis between omega-3 and omega-6) is not well explained. The term immunomodulation is referring precisely to this mechanism.

A20: Revised to show that in the presence of aspirin and subsequent acetylation of the COX-2 enzyme, production of resolvins is enhanced thereby reducing the impact of inflammatory prostaglandins generated from arachidonic acid. 

Q21:Row 221: What is the link between cancer cell proliferation and inflammation?

A21: Chronic Inflammatory and hypoxic conditions are known to provide a conducive environment for malignant cells to proliferate, exhaust and evade immune cells. Reduction in proliferation reduces tumor progression. 

Q22:Row 222: How can DHA and EPA have an anti-inflammatory effect by increasing the level of MHC I on cell surface? Rather, it is expected to trigger an inflammatory response?! Please explain!

A22: Revised to provide further clarification. Efficient antigen presentation via MHC I leads to faster immune resolution and reduction of a chronic inflammatory environment..  

Q23: Row 226: This paragraph speaks almost exclusively of the effects of DHA. What are the effects of AA (omega-6)?

A23:Revised to include the role of AA and highlight the importance of aν optimal balance between ω-3  to ω-6. 

Q24: Table 1: As the review talks about polyunsaturated acids, it would be useful to subdivide Table 1 into oils rich in omega 3 and 6.

A24: We have specified this in Table 1.

Q25:Table 1: Nutritionally, flaxseed oil and linseed oil are the same. In my opinion, it is not necessary to present the fatty acid content of both oils in the table. Flaxseed oil could be replaced advantageously by walnut oil, another oil rich in omega 3, as mentioned on row 158.

A25: We have deleted the linseed oil and added walnut oil in Table 1. 

Q26:Table 2: It may be useful to differentiate COX-1 from inducible COX-2

A26:Revised as per reviewer.

Round 2

Reviewer 1 Report

The authors have responded to the previous concerns and the quality of this manuscript has been greatly improved. The current revised manuscript is acceptable for publication.

Author Response

Thank you!

Reviewer 3 Report

General comment:

The message of the manuscript has been significantly improved. However, there is still no mention of oleic acid in the simple summary, the abstract and the conclusions. The authors should either mention the effects of oleic acid in these sections or remove all mention of monounsaturated fatty acids (oleic acid) in their manuscript and put back in the title "polyunsaturated fatty acids".

Detailed comments:

Row 73: “high levels of MDA may initiate ROS proliferation and a massive oxidative catastrophe” It should be said here more clearly that PUFAs can also have a pro-oxidant effect when their peroxidation products, such as MDA, accumulate in cells and cause DNA damage.

Row 131: “that a particular a diet can provide protection for a particular disease”. Please, correct!

Rows 132, 163, 165, 171, 187, 202: “Polyunsaturated fatty acids” are already mentioned in row 40 and should be abbreviated everywhere else.

Rows 217, 232, 251, 252: “Arachidonic acid” is already mentioned in row 141 and should be abbreviated everywhere else.

Row 275: TNFalpha instead of TNFa

Author Response

The message of the manuscript has been significantly improved. However, there is still no mention of oleic acid in the simple summary, the abstract and the conclusions. The authors should either mention the effects of oleic acid in these sections or remove all mention of monounsaturated fatty acids (oleic acid) in their manuscript and put back in the title "polyunsaturated fatty acids".

MUFAs and oleic acid have been included in abstract, conclusion and sections.

Detailed comments:

Row 73: “high levels of MDA may initiate ROS proliferation and a massive oxidative catastrophe” It should be said here more clearly that PUFAs can also have a pro-oxidant effect when their peroxidation products, such as MDA, accumulate in cells and cause DNA damage.

Revised as suggested.

Row 131: “that a particular a diet can provide protection for a particular disease”. Please, correct!

Corrected.

Rows 132, 163, 165, 171, 187, 202: “Polyunsaturated fatty acids” are already mentioned in row 40 and should be abbreviated everywhere else.

Revised.

Rows 217, 232, 251, 252: “Arachidonic acid” is already mentioned in row 141 and should be abbreviated everywhere else.

Revised.

Row 275: TNFalpha instead of TNFa

Revised to greek letter